# Preclinical Efficacy and Toxicity Analysis of the Pan-Histone Deacetylase Inhibitor Gossypol for the Therapy of Colorectal Cancer or Hepatocellular Carcinoma

**DOI:** 10.3390/ph15040438

**Published:** 2022-04-01

**Authors:** Mascha Mayer, Alexander Berger, Christian Leischner, Olga Renner, Markus Burkard, Alexander Böcker, Seema Noor, Timo Weiland, Thomas S. Weiss, Christian Busch, Ulrich M. Lauer, Stephan C. Bischoff, Sascha Venturelli

**Affiliations:** 1Institute of Nutritional Medicine and Prevention, University of Hohenheim, 70599 Stuttgart, Germany; mascha.mayer@icloud.com; 2Department of Internal Medicine VIII, University Hospital Tuebingen, 72076 Tuebingen, Germany; alex.c.berger@gmx.de (A.B.); timo.weiland@gmx.de (T.W.); ulrich.lauer@uni-tuebingen.de (U.M.L.); 3Department of Nutritional Biochemistry, Institute of Nutritional Sciences, University of Hohenheim, 70599 Stuttgart, Germany; christian.leischner@uni-hohenheim.de (C.L.); olga.renner@uni-hohenheim.de (O.R.); markus.burkard@uni-hohenheim.de (M.B.); 4Evotec SE, 22419 Hamburg, Germany; aleboe@web.de; 5Department of Dermatology, Eberhard Karls University of Tuebingen, 72076 Tuebingen, Germany; seema.noor@med.uni-tuebingen.de; 6Center for Liver Cell Research, Children’s University Hospital (KUNO), University Hospital Regensburg, 93042 Regensburg, Germany; thomas.weiss@ukr.de; 7Dermatologie zum Delfin, 8400 Winterthur, Switzerland; ch_busch@hotmail.com; 8German Cancer Consortium (DKTK), DKFZ Partner Site, 72076 Tuebingen, Germany; 9Department of Vegetative and Clinical Physiology, Institute of Physiology, University of Tuebingen, 72074 Tuebingen, Germany

**Keywords:** gossypol, histone deacetylase, epigenetics, AT-101, colon cancer, liver cancer

## Abstract

Gossypol, a sesquiterpenoid found in cotton seeds, exerts anticancer effects on several tumor entities due to inhibition of DNA synthesis and other mechanisms. In clinical oncology, histone deacetylase inhibitors (HDACi) are applied as anticancer compounds. In this study, we examined whether gossypol harbors HDAC inhibiting activity. In vitro analyses showed that gossypol inhibited class I, II, and IV HDAC, displaying the capability to laterally interact with the respective catalytic center and is, therefore, classified as a pan-HDAC inhibitor. Next, we studied the effects of gossypol on human-derived hepatoma (HepG2) and colon carcinoma (HCT-116) cell lines and found that gossypol induced hyperacetylation of histone protein H3 and/or tubulin within 6 h. Furthermore, incubation with different concentrations of gossypol (5–50 µM) over a time period of 96 h led to a prominent reduction in cellular viability and proliferation of hepatoma (HepG2, Hep3B) and colon carcinoma (HCT-116, HT-29) cells. In-depth analysis of underlying mechanisms showed that gossypol induced apoptosis via caspase activation. For pre-clinical evaluation, toxicity analyses showed toxic effects of gossypol in vitro toward non-malignant primary hepatocytes (PHH), the colon-derived fibroblast cell line CCD-18Co, and the intestinal epithelial cell line CCD 841 CoN at concentrations of ≥5 µM, and embryotoxicity in chicken embryos at ≥2.5 µM. In conclusion, the pronounced inhibitory capacity of gossypol on cancer cells was characterized, and pan-HDACi activity was detected in silico, in vitro, by inhibiting individual HDAC isoenzymes, and on protein level by determining histone acetylation. However, for clinical application, further chemical optimization is required to decrease cellular toxicity.

## 1. Introduction

The effects of bioactive components from cotton seeds have received significant attention in the 1960s, as a subnormal fertility was found in Chinese Henan and Hubei provinces, which was linked to the use of crude cotton seed oil [1]. As a non-steroidal male contraceptive, gossypol was then isolated from cotton seeds (*Gossypium hirsutum*) and has been tested in clinical trials as antifertility agent with an overall efficacy of 99.89% [2]. Gossypol, 7-[8-formyl-1,6,7-trihydroxy-3-methyl-5-(propan-2-yl)naphthalen-2-yl]-2,3,8-trihydroxy-6-methyl-4-(propan-2-yl)naphthalene-1-carbaldehyde (https://pubchem.ncbi.nlm.nih.gov/compound/3503, accessed on 29 March 2022), is a yellow-colored complex polyphenol found in the small intercellular pigment glands in cotton leaves, stems, roots, and seeds, and is a well-studied secondary plant metabolite from glanded seeds [3,4].

In the meantime, numerous studies have demonstrated the bioactive functions of this plant-derived compound [4,5,6] comprising antifertility, antiviral, antimicrobial, and antioxidative capacities [7,8,9]. Anti-proliferative, anti-metastatic, and pro-apoptotic effects of gossypol were described for several human malignancies, including colon, pancreatic, adrenal, prostate, and breast cancers, as well as glioma and leukemia [10,11,12,13,14,15,16,17,18]. These observations have generated intensive interest in the biomedical field, and large amounts of research efforts have been directed at understanding the potential of pharmacological applications [8].

Natural gossypol consists of (+)- and (−)-enantiomers, and the ratio of those varies in different plants species [8,9]. Gossypol exhibits atropisomerism, a special case of enantiomerism, in which the rotation around the binaphthalene single bond is restricted by the sterically demanding substituents. Analysis of the enantiomeric relations has shown an excess of the (+)-enantiomer in most cotton seeds and in *Thespesia populnea*, with *Gossypium barbadense* being the main variety with an excess of the (−)-enantiomer. In addition, depending on the solvent, gossypol exists in three symmetrical or asymmetrical tautomeric forms: aldehyde, lactol, and ketol. The presence of the phenolic hydroxyl and aldehyde groups defines its biological reactivity. Under physiological conditions, there is an equilibrium between the aldehyde and lactol tautomer. Initially, in 1984, gossypol was found to have anti-tumor effects against several tumor cell lines grown in tissue culture [19]. Thereby, AT-101, the R(−)-enantiomer of gossypol, possessed a higher antitumor activity than the (+)-enantiomer or racemic gossypol [8,9].

Histone deacetylases (HDACs) are key players of transcriptional repression [20]. They remove acetyl groups from DNA-binding histone proteins and decrease chromatin accessibility for transcription factors and generate specific, repressive effects on gene expression [21]. To date, there are 18 known HDACs that are grouped into two families based on the presence of a conserved deacetylase domain and their dependence on specific cofactors: the histone deacetylase family and the sirtuin protein family. The histone deacetylase family is subdivided into class I (HDAC1, 2, 3, and 8), class II (HDAC4, 5, 6, 7, 9, and 10), and class IV (HDAC11) based on sequence similarity to yeast deacetylases, which are zinc-dependent amidohydrolases and are mostly localized in the nucleus with ubiquitous distribution throughout human cells and tissues [22]. The sirtuin proteins are classified within the class III HDACs, which require nicotinamide adenine dinucleotide (NAD) as a cofactor for their catalytic function [23]. In addition to histones, HDACs also deacetylate a variety of other non-histone proteins, including transcription factors [24,25,26] and other cellular proteins controlling cell growth, differentiation, and apoptosis [27]. The aberrant expression of HDACs causes changes in the transcription of key genes regulating important cancer pathways, such as cell proliferation, cell cycle regulation, and apoptosis [28,29]. Therefore, HDACs are considered as potential therapeutic targets, and the development of HDAC inhibitors (HDACi), antagonizing tumor initiation and/or growth, is a promising topic in the search for new anticancer drugs [28].

Recently, a growing number of nutritional compounds have been identified exhibiting epigenetic activity, which is able to influence gene expression by modifications of the DNA or DNA-associated proteins without changing the DNA sequence itself [30]. Therefore, an innovative therapy strategy is to use synthetic or naturally occurring HDACi as anticancer drugs [31,32,33,34,35]. Today, some HDACi, such as vorinostat (suberoylanilide hydroxamic acid (SAHA), Zolinza^®^ (Merck Sharp & Dohme Corp., Kenilworth, NJ, USA)), belinostat (PXD101, Beleodaq^®^ (Spectrum Pharmaceuticals, Inc., Irvine, CA, USA)), the microbial metabolite romidepsin (FK288, Istodax^®^ (Celgene Corporation, Summit, NJ, USA)), tucidinostat (chidamide, Epidaza^®^ (Shenzhen Chipscreen Biosciences Co., Ltd., Shenzhen, China)), panobinostat (LBH589, Farydak^®^ (Secura Bio, Inc., Las Vegas, NV, USA)), and valproic acid (VPA) have been approved by the FDA for the treatment of cancer [29,36,37,38,39]. Over ten different HDACi are currently in use for clinical trials, and more are being investigated to augment the efficacy of chemotherapy, immunotherapy/cancer vaccines, and to overcome multi-drug resistance, which is common to diverse cytostatic or receptor-mediated drugs [40]. The nutraceutical screening for HDACi by Mazzio et al. identified gossypol as a non-fermented phytochemical with a HDACi activity with an IC_50_ < 200 μg/mL [40]. Elevated levels of several HDACs have been reported in colon cancer [41] and hepatocellular cancer [42,43,44], while downregulation of specific HDACs inhibits growth of cancer cells in vitro and intestinal tumorigenesis in vivo [45,46]. Recently, Renner et al. systematically reviewed the efficacy of gossypol/AT-101 in clinical cancer trials [47]. It was shown that this drug candidate exhibits promising effects in distinct subgroups of patients, supporting further specification of AT-101-sensitive cancers, as well as the establishment of an effective AT-101-based therapy and the requirement for precise toxicity profiling.

In this study, we aimed to explore the value of gossypol as potential HDACi and anticancer drug. The HDACi activity of gossypol was characterized in detail using in silico docking analysis for human class I and II HDAC enzymes and was confirmed in a specific HDAC inhibition in vitro assay. Additionally, the acetylation status of histone H3 was evaluated. The anti-proliferative and apoptotic effects were assessed in vitro on human hepatoma and colon carcinoma cell lines. Finally, the toxicity profile was determined on human non-malignant cells from tissue that corresponds to the investigated cancer entities and in vivo in a chicken embryotoxicity assay.

## 2. Results

### 2.1. Anti-Proliferative Effects of Gossypol on Human Hepatoma and Colon Carcinoma Cell Lines

Since gossypol exerts anti-proliferative effects on different tumor entities [48,49], we investigated the therapeutic potential of gossypol on different human hepatoma and colon cancer cells lines. To obtain more information about a possible p53 dependency, p53 wild-type (wt), as well as p53 aberrant cell lines were used (HepG2 and HCT-116 are p53 wt, Hep3B has a p53 gene deletion, HT-29 has a R273H mutated p53 gene locus) [50,51,52,53]. Proliferation (Figure 1a) and cellular viability (Figure 1b,c) of hepatoma (HepG2, Hep3B) and colon carcinoma (HCT-116 and HCT-29) cell lines were subsequently monitored over a time period of up to 106 h.

Measurement of the cellular impedance, represented by the cell index (CI), reflects the cellular status. Hence, an increase in CI usually displays cellular growth, whereas a reduction indicates perturbations of cell viability [54,55]. After 10 h, CI was normalized and cells were treated once with increasing concentrations of gossypol (5 µM, 10 µM, 20 µM, or 50 µM) and incubated for additional 96 h. A concentration of 0.1% Triton X-100 served as positive control. In comparison to the medium, which was defined as negative treatment control, normalized CI of all cell lines showed a concentration-dependent decline following application of gossypol (Figure 1a). At first glance, there was a different curve progression between the analyzed cell lines. A concentration-dependent flattening of the curves depicted as mean CI values was already achieved with 5 µM gossypol in HepG2 and Hep3B cells after 42 h. In contrast, HCT-116 and HT-29 cells showed only a slight reduction in the respective CI values at concentrations of 5 µM and 10 µM of gossypol in comparison to untreated control cells at the beginning of the treatment period, while concentrations of gossypol ≥ 20 µM were able to strongly interrupt the exponential increase in CI, thereby efficiently inhibiting tumor cell proliferation.

As changes in CI values can be caused by several reasons, such as alteration of cellular morphology, senescence, detachment of cells, or different cell death mechanisms, the anti-proliferative effects of gossypol were further analyzed via sulforhodamine B (SRB) assay. Cell viability of gossypol-treated tumor cells (5 µM, 10 µM, 20 µM, or 50 µM) was compared to untreated controls (Figure 1b,c). In line with the real-time proliferation experiments, the overall decrease in protein levels detected by the SRB assay in HepG2, Hep3B, HCT-116, and HT-29 cells confirmed the results of the real-time cell monitoring. As shown in Figure 1b,c (left panel), 5 µM gossypol decreased cell viability below 50% in the human hepatoma cell lines after 48 h, and even below 25–30% after 96 h. Notably, viability of cells was completely abolished after 48 h treatment with 50 µM gossypol, and after 96 h with 20 µM, independent of the p53 status of the hepatoma cells. p53 wt colon carcinoma cells HCT-116 were more susceptible to treatment with 5 µM compared to p53 mutated HT-29 cells (Figure 1b,c, right panel). Thereby, after 96 h incubation with 5 µM gossypol, cell viability was reduced to approximately 25–30% for both hepatoma cells and the p53 wt colon carcinoma cell line HCT-116. p53 mutated HT-29 cells displayed a diminished decrease in cell viability (about 10% decrease after 48 h and 20% after 96 h) after treatment with a lower gossypol concentration of 5 µM compared to the other cell lines tested. However, a total suppression of cell viability in HT-29 cells was observed with 50 µM gossypol after 96 h. According to these results, the strongest anti-proliferative effect of gossypol was achieved 48 h after treatment with 50 µM, with human-derived hepatoma cells being the most susceptible cell lines, which showed a p53-independent rapid reduction in viability.

### 2.2. Gossypol Induces Apoptosis in Human Hepatoma and Colon Cancer Cell Lines

Due to the pronounced suppression of cell proliferation by gossypol described above, we next analyzed the underlying cell death mechanism by FACS analysis. Therefore, the sub2N population, as a marker for apoptotic cell death, was observed over time. Determination of sub2N population showed that gossypol induced a dose-dependent increase in sub2N events from 6 h to 72 h in all cell lines tested (Figure 2). HepG2 cells exhibited the highest basal apoptosis rate among the investigated cell lines. In addition, p53 wt cell lines (HepG2 and HCT-116) were more susceptible for apoptosis induction by gossypol. A significant increase in subN2 events was achieved with ≥10 µM gossypol for HepG2, Hep3B, HCT-116, and HT-29. In line with the findings from cell impedance and viability analyses, 50 µM gossypol induced a strong elevation of subN2 events in wt cell lines (>50% compared to controls). This was also observed for Hep3B and HT-29 but was less pronounced. The induction of necrosis by gossypol was excluded due to negative Annexin V/propidium iodide (PI) double staining (data not shown).

### 2.3. Characterization of Gossypol-Induced Apoptosis via Caspase Assay

Caspase activation is a feature of apoptotic cell death and usually occurs before the loss of membrane integrity and the appearance of sub2N DNA. Upon induction of apoptosis, cell degradation is initiated by special cysteine proteases, the so-called caspases, which are intracellularly localized as inactive procaspases [56]. The activation of caspases results in degradation of nuclear membrane lamins and actin in the cytoskeleton, and in the deactivation of repair enzymes [57]. For both p53 wt cell lines (HepG2 and HCT-116), the possible induction of caspase 3/7 proteases after gossypol application was analyzed (Figure 3). As a positive control for apoptosis induction, 5 µM staurosporine (STS), a prototypical ATP-competitive kinase inhibitor with high binding affinity and little selectivity [58], was used to induce caspase activity and to demonstrate the validity of the assay. During a time period of 48 h, the measurements were performed every 12 h. Caspase 3/7 activity is shown as fold change rate compared to the untreated control. The increase in caspase 3/7 activity was already detectable after treatment with 50 µM gossypol and 12 h incubation for both cell types. For this concentration, the strongest inductions (8.46- and 5.93-fold) were measured after 24 h for HepG2 and HCT-116, respectively. Incubation with 20 µM gossypol for 36 h was also able to stimulate caspases activity. However, this effect was weaker (2.15-fold in HepG2 and 2.72-fold in HCT-116 cells) compared with those values of 50 µM treatment for 24 h. Longer incubation (36 h or 48 h) with 50 µM gossypol resulted in a decline of caspase activity (HepG2 3.54- and 2.32-fold, HCT-116 1.03- and 0.84-fold compared to control), which is explained by the depletion of available caspase enzymes, and thus, by advanced apoptosis. For 5 µM, no induction of caspases was seen at the observed time points in both cell lines, which is in line with sub2N determination also showing no increase at this concentration.

### 2.4. Gossypol-Related ATP Depletion in Human Hepatoma and Colon Cancer Cell Lines

Another examination method for the detection of cell damage is the determination of the cellular ATP content. ATP is present in all metabolically active cells and serves as a marker for cell viability. In case of necrosis and apoptosis, its concentration decreases after an initial rise. The aim of the following experiments was to identify the concentrations of gossypol, which resulted in decreased and/or elevated ATP levels. Overall, a time- and concentration-dependence was observed for all four tested cell lines after treatment with gossypol (Figure 4). In HepG2 and Hep3B cells, an initial, statistically significant increase in ATP content was observed with 10 µM at the first time point (6 h). Higher concentration of 20 µM exhibited a more pronounced elevation in ATP content at 6 h. After 6 h, 20 µM gossypol induced the highest ATP concentration in HTC-116 and HT-29 cells (328% and 337%, respectively, compared to control). Remarkably, in three cells lines (with the exception of HepG2) the ATP level induced with 50 µM gossypol was lower than that of 20 µM (86% for Hep3B (for 50 µM 155%), 204% for HCT-116, and 160% for HT-29). This fact could be explained by the strong apoptosis-inducing effect of gossypol. In this regard, the highest increase in ATP concentration after 50 µM treatment seemed to occur before the 6 h measurement point and then led to a continuous decrease through all further time points (12 h, 24 h, 48 h). Already after 24 h incubation with 50 µM gossypol, a decrease in ATP below 50% was consistently detected for all cell lines. Furthermore, the cell lines showed that lower gossypol concentrations also contributed to an initial rise in ATP, but this required a longer incubation period with the compound. Regardless of the p53 status, all cell lines responded to gossypol treatment with an initial increase in ATP levels followed by ATP depletion after 48 h treatment, resulting in metabolic inactivation. Thereby, p53 wt cells seem to be more sensitive to gossypol than cells with mutated p53.

### 2.5. In Vitro Analysis and In Silico Screening of pan-HDACi Activities of Gossypol

Based on the apoptosis- and cell-viability-induced effects of gossypol described above, we analyzed the HDAC inhibitory effect on the individual classical human HDACs using a cell-free HDACi screening assay. A concentration of 50 μM was chosen to ensure comparability with prior experiments. The profile analysis indicated an inhibition by gossypol for all HDACs ranging from 56.0 to 82.7% (Table 1).

Further assessment of HDAC enzyme inhibition (HDAC class I (HDAC2, HDAC8) and class II (HDAC4, HDAC7)) by gossypol was performed by in silico docking analysis (Figure 5a). Protein data allowing in silico docking of class IV HDAC enzymes were not available at the time point of analysis. There are two conditions, which have to be met for the identification of an HDACi: the structural property to fit into the binding pocket of HDAC enzymes and the capacity to interact with key interaction points, such as the zinc ion in the catalytic center. The established HDACi trichostatin A (TSA) served as reference inhibitor. For direct comparison of the docking results, the GoldScore was used to measure the binding potential. Considering these in silico data, gossypol seemed not to directly fit into the binding pocket. For this reason, a new docking run without constraints, using only HDAC8 and gossypol, was performed. Therefore, the goal was to find the spatial position on the HDAC enzyme with which gossypol would most strongly, and thus most likely, interact. The position obtained by this run confirmed the previous finding that gossypol did not enter the binding pocket as required for a competitive inhibitor in contrast to TSA [59] but interacted slightly laterally from the binding pocket with the HDAC.

To declare gossypol a pan-HDAC inhibitor, further analysis on cellular level was performed. Treatment of cells with an HDACi typically results in hyperacetylation of histone proteins. Therefore, the amounts of acetylated histone protein H3 (ac-H3) and acetylated tubulin (ac-tubulin) were analyzed within the most susceptible p53 wt cell lines HepG2 and HCT-116 (Figure 5b). After 6 h treatment with 5 μM, 10 μM, 20 μM, or 50 μM of gossypol, increased levels of ac-H3 and ac-tubulin were detected. The clinically established HDACi vorinostat (SAHA) served as positive control. Comparing the Western blot results of gossypol with SAHA, gossypol caused only a moderate increase in both acetylated proteins.

### 2.6. Toxicity Profile of Gossypol on Non-Malignant and Embryonic Cells

For pre-clinical evaluation, toxicity studies on non-malignant cells are of great importance. Therefore, the effect of gossypol was tested on primary human hepatocytes (PHHs) and colon-derived non-malignant cells (CCD-18Co, CCD 841 CoN). Incubation for 48 h and 96 h with gossypol (5 μM, 10 μM, 20 μM, 50 μM) and determination of cell viability via SRB assay showed a concentration- and time-dependent reduction in cell viability (Figure 6a). A significant decrease in viability was already initiated with 5 μM gossypol for both cell types (PHH and CCD-18Co) after 48 h. Therefrom, the breakdown of viability of these cells already started at the lowest concentration (5 μM) and was further pronounced with higher concentrations. A concentration of 50 μM gossypol reduced the viability of PHHs and CCD-18Co to levels lower than 50% and 15%, respectively. In CCD 841 cells, a significant decrease in viability was shown only with 50 μM after 48 h incubation period. This cell line was more resistant than PHHs and CCD-18Co cells to lower gossypol concentrations for 48 h treatment duration. A longer incubation period (96 h) led to a significant reduction in viability already with 10 μM of substance’s administration.

Lactate dehydrogenase (LDH) is a widely used marker for membrane integrity that is relatively stable under cell culture conditions after it is released into the supernatant by dying cells. To evaluate cellular damage, we investigated the release of LDH over time (24 h, 48 h, 72 h, 96 h) after treatment with different gossypol concentrations (5 µM, 10 µM, 20 µM, 50 µM). LDH release was already significantly increased at 20 µM and 50 µM after 24 h (Figure 6b).

Aspartate aminotransferase (AST) is expressed in many tissues of the body, but especially in liver and muscle cells. This enzyme is mainly localized in the cytosol and mitochondria of the cell and is, therefore, released into the medium even in the case of minor cell damage. In the case of more severe damage, AST is also released from the mitochondria. As a marker for cellular damage and integrity, we also tested the release of AST into the supernatant at different time points (24 h, 48 h, 72 h, 96 h) (Figure 6c). In line with results from the SRB test, concentration-related elevation of cell damage was initiated by gossypol already after 24 h. A significant increase was noted beginning from 20 μM gossypol. Remarkably, the effect of 50 μM gossypol and 24 h incubation was comparable to Triton treatment, achieving a 10-fold AST release compared to control. Longer incubation time led to a reduction in AST levels, probably due to pronounced cell death and degradation of AST in the supernatant.

The strong cytotoxic effect of gossypol in vitro on non-malignant cell lines suggests that this dimeric sesquiterpene aldehyde may also have a strong cytotoxic effect in vivo with possible teratogenicity. Consequently, the effect of gossypol on incubated chicken embryos was investigated. The determination of embryotoxicity revealed that gossypol caused a high mortality among the chicken embryos (Figure 6d). Concentrations of 20 μM and 50 μM gossypol showed a 100% death rate within the first 24 h. The other concentrations studied in the previous assays, 5 μM and 10 μM, reduced the probability of survival within 72 h. Lower concentrations (1 µM and 2.5 μM) that were not used in the previous experiments resulted in no death of embryos (1 μM), whereas 2.5 μM resulted in the death of one embryo after 24 h and of another after 48 h. The LD_50_ of gossypol was determined to be approximately 4.5 μM.

## 3. Discussion

Some HDACi were already tested in clinical trials and are approved by the FDA for the treatment of cancer [60,61]. In this study, we showed that gossypol exhibits pan-HDAC inhibitory activity that causes antiproliferative and apoptotic events. Remarkably, these effects were induced in human hepatoma and colon carcinoma cell lines, as well as in non-malignant human primary liver and colon cells. In addition, gossypol had a toxic impact on non-malignant cells and was embryotoxic at low concentrations.

### 3.1. Apoptosis

In general, there are two fundamental mechanisms by which HDAC regulate gene expression, inducing hyperacetylation of histone and non-histone proteins [36,62]. Hyperacetylation of histones changes the expression of epigenetically repressed genes in tumor cells, causing the induction of apoptosis [36]. The hyperacetylation of the transcriptional factors and chaperone proteins, e.g., tumor-suppressing transcriptional factor p53, results in enhanced transcriptional activity, altering target gene expression and leading to tumor suppression [36]. In colon cancer cells, the impact of this effect varies between the analyzed cell lines, separating sensitive and resistant lines. In HDACi sensitive cells, apoptotic or transcriptional response by reactivation of expression of immediate early and stress response genes could be observed [63]. In addition, across tumor types, this apoptotic sensitivity also depends on the pro-apoptotic function of the activating transcription factor 3 (ATF3) operating through direct transcriptional repression of the pro-survival factor *BCL-X_L_* [64].

Loss of the p53 tumor suppressor pathway contributes to the development of most human cancers [65]. The tumor suppressor gene *TP53* is mutated in ~50% of human cancers, and therefrom, derived protein p53 plays a major role in the response of malignant as well as non-transformed cells to many anticancer therapeutics, particularly those that cause DNA damage [66]. Gossypol induces DNA damage and activates p53 [67]. Inhibited cell growth, apoptosis, and autophagy through p53-dependent pathways were demonstrated for gastric cancer [68]. Additionally, gossypol-induced apoptosis is related to p53 mutation status [69]. In a mouse medulloblastoma model, wt p53 exhibited a much more robust induction of apoptosis in response to gossypol treatment compared to tumor cells with mutated p53. In contrast, gossypol evoked cancer cell apoptosis regardless of p53 status in human breast cancer cells [70]. To obtain more information about a possible p53 dependency, we analyzed p53 wt, as well as p53 mutated hepatoma and colon carcinoma cell lines [50,51,52,53]. In line with the findings of Xiong et al. [70], no significant differences were observed for the tested cell lines with respect to their p53 genetic background.

Gossypol-induced apoptosis occurred via both caspase-dependent and -independent pathways [8]. Our data confirm the growing caspase 3/7 activity in a time- and concentration-dependent manner for p53 wt immortalized hepatoma and colon carcinoma cells (Figure 3).

### 3.2. HDACi Docking Analysis

The classical way to inhibit an HDAC is based on modeling of a HDAC pharmacophore and requires three elements (zinc binding group, linker, and cap) [61,71,72]. However, there are also molecules that untypically bind with high affinity to HDACs outside the active pore, such as tasquinimod [73]. Our in silico prediction analysis for gossypol binding properties showed the docking more laterally to the active site of HDACs classes I and II (HDAC 2, 4, 7, and 8). Allosteric modulators as dual mechanism agents, covalent inhibitors and non-active site modulators need not adhere to the standard model for HDAC active site binding, and therefore, the implementation of unconventional approaches for HDAC definition was suggested for accurate predictions [71], which might also be applicable to gossypol. The gossypol-related suppression of HDAC activity in a standardized HeLa nuclear extract assay, as demonstrated by Mazzio [40], underpins our findings from the HDACi screening analysis (Table 1). Further, the predicted inhibitory potential was verified according to the acetylation potential of gossypol by Western blot analysis. In contrast to the clinically used HDAC inhibitor SAHA or other pan-HDACi, such as kaempferol or resveratrol [34,35,74,75], our results show that an increase in acetylated H3 due to gossypol treatment was moderate and partially concentration dependent in both cell lines. Remarkably, in HepG2 cells, a concentration-related increase in acetylated tubulin was clearly detectable but not in HCT-116 cells, which indicates that the epigenetic modifications in this cell model might at least be partly based on HDAC inhibition (Figure 5b). Of note, tumor cells show an imbalance in the regulation of HDACs, mostly due to an overexpression of one or more HDAC enzymes [76]. The upregulation of even a single HDAC enzyme can induce oncogenic cell transformation and tumorigenesis [77]. In summary, according to our in vitro and in silico analyses, we suggest gossypol as a pan-HDACi. The observed HDACi potential of gossypol therefore supports the hypothesis that nutritional HDACi exert moderate inhibitory activities [78].

### 3.3. Mitochondria

Another intracellular target where gossypol has multiple modes of action are mitochondria. Mitochondriotropic derivatives of phenolic compounds, such as quercetin and resveratrol, efficiently induced cancer cell death in vitro [79]. Treatment with gossypol selectively increased hydrogen peroxide levels and impaired mitochondrial respiration [80]. Moreover, gossypol caused depletion of ATP via uncoupling of the oxidative phosphorylation from the electron transport [81] or served as ion channel blocker regulating volume-sensitive osmolyte channels and altering the activity of ATP-regulated transporters by reducing cellular ATP [82]. The detection of cell damage via determination of cellular ATP is an established method for metabolically active cells. According to our data, a rise in ATP levels following incubation with gossypol occurred in a concentration- and time-related manner. Generally, colon carcinoma cells lines seemed to respond to gossypol with higher increase in relative ATP content (maximum ≥ 3-fold, compared to control) compared to hepatoma cells (maximum ≥ 2-fold, compared to control). However, independent of tissue origin, both p53 wt cells lines were more sensitive to gossypol treatment and exhibited more significant effects. A pronounced gossypol-induced increase (already after 6 h) and then a subsequent decline of ATP (after 48 h) in all analyzed cell lines suggested a very early initiation of cell damage depending on the concentration of the substance. This early initiated cell damage might explain the missing or weak ac-H3 immunoblotting signal of HepG2 cells for 50 μM of gossypol due to fast and massive cell death. These findings also corroborate the results from measurements of cellular impedance via real-time proliferation, in which 50 μM of gossypol evoked a rapid decline (few hours after gossypol treatment) of CI values.

### 3.4. Toxicity and Comparison of HDACs

Considering a possible application of gossypol in patients with hepatoma or colon carcinoma, we tested the influence of gossypol on non-malignant cell lines. For hepatoma, non-malignant PHHs from three different donors, and for colon carcinoma, the non-malignant cell lines CCD-841 CoN (epithelial) and CCD-18Co (fibroblast), were used. The application of gossypol on PHHs and CCDs resulted in a decrease in viability. The toxic effect of gossypol on liver has already been described for oral and intravenous application in animals [83,84] and in rat liver cell lines [85,86]. By contrast, in clinical trials, gossypol did not induce toxic side effects in vivo and was considered as clinically safe at low concentrations [13,87]. To the best of our knowledge, the cytotoxic effect of gossypol on PHHs was shown for the first time in this study. The non-malignant colon cell lines tolerated gossypol after 48 h only up to a concentration of 20 µM and showed strong decrease in viability at higher concentrations. Lower concentrations of gossypol also exhibited toxicity, but after a longer treatment period. The observed toxic effects of gossypol on the non-malignant cells could be due to various reasons. Nevertheless, we suppose that the detected cytotoxic effects of gossypol were only partially based on the newly discovered pan-HDACi activity. In the present study, chicken embryos were incubated with gossypol for 24 h, 48 h, 72 h, or 96 h. The results show, for the first time, a severe embryotoxicity, even in the low micromolar range of ≥2.5 µM. A topical application of gossypol without systemic absorption might be an alternative treatment option.

Together, the results of our study provide further evidence for the anti-proliferative and pro-apoptotic effects of gossypol on human hepatoma and colon carcinoma cell lines and on non-malignant human primary liver and colon cells. The profiling for all human HDACs showed an HDAC-inhibitory potential of gossypol of 50% or higher, although in silico analysis suggests that gossypol interacts slightly laterally within the binding domain (Figure 5a). The current study was conducted on a selected set of liver and colon cancer cell lines with specific genetic p53 backgrounds. It would be valuable to confirm our findings in a larger set of liver and colon tumor cell lines and in patient-derived primary tumor cells. Future studies on digestion-related modifications of gossypol, the degradation in the gastrointestinal tract, and the effects of its metabolites could provide further insight into the bioavailability and therapeutic efficacy of gossypol.

## 4. Materials and Methods

### 4.1. Ethics Statement

Non-malignant PHHs from different donors were provided from the charitable state-controlled foundation Human Tissue & Cell Research (HTCR; http://www.htcr.de, accessed on 29 March 2022). Written informed patient’s consent, approved by the local ethical committee of the University of Regensburg, Germany [88,89], was obtained prior to sample collection. All experiments involving human tissues and cells were carried out in accordance with The Code of Ethics of the World Medical Association (Declaration of Helsinki).

### 4.2. Cell Culture and Reagents

Human-derived hepatic cancer cell lines HepG2 and Hep3B (DSMZ numbers: ACC 180 and ACC 93) were obtained from the German Collection of Microorganisms and Cell Cultures (Deutsche Sammlung von Mikroorganismen und Zellkulturen (DSMZ), Braunschweig, Germany). Human colon carcinoma cell lines HT-29 and HCT-116 (ATCC^®^-Numbers: HTB-38™ and CCL-247™) were obtained from LGC Standards (Wesel, Germany). All cell lines were grown as monolayer and cultured routinely in 175 cm^2^ polystyrene flasks (Corning Life Sciences, Tewksbury, MA, USA) at 37 °C, in a humidified atmosphere with 5% (*v*/*v*) CO_2_. HepG2, Hep3B, HCT-116, and HT-29 cell lines were cultivated in Dulbecco’s Modified Eagle Medium (DMEM) (Biochrom, Berlin, Germany) with 10% heat inactivated fetal calf serum (FCS) (Gibco^®^ FBS, Invitrogen, Darmstadt, Germany) and 2 mM L-glutamine (Life Technologies, Rockville, MD, USA).

PHHs were provided by T.S. Weiss and cultured as described [88], briefly, in DMEM with 100 U/mL penicillin and streptomycin (Serva, Heidelberg, Germany), 2 mM L-glutamine (Life Technologies), 18.8 µg/mL hydrocortisone (Merck, Darmstadt, Germany), and 1.68 mU/mL insulin (Novo Nordisk, Bagsvaerd, Denmark).

Non-malignant colon cell lines CCD 841 CoN (epithelial) and CCD-18Co (fibroblast) were purchased from American Type Culture Collection (ATCC, Manassas, VA, USA) and cultured in DMEM with 10% FCS.

(±)-Gossypol from cotton seeds was obtained from Sigma-Aldrich (Taufkirchen, Germany; ≥95% purity, CAS RN 303-45-7), solved in Dimethylsulfoxide (DMSO) (CAS: D5879, Sigma-Aldrich) as a stock solution of 100 mM and stored at −20 °C. The final dilutions were made immediately before use. The DMSO concentration in the assay did not exceed 0.1% (*v*/*v*) and was not cytotoxic to the cells.

### 4.3. Docking Analysis

Docking analysis was performed with human HDAC2, 4, 7, and 8 with gossypol and the reference HDACi TSA. Due to the acidity of the phenol groups present in gossypol, the non-ionized and different deprotonated isoforms were considered. All ligands were prepared using the molecular operation environment (MOE, version 2009.10, Chemical Computing Group Inc., Montreal, QC, Canada). Three-dimensional representations of the ligands were obtained by energy minimization (Rebuild3D function with preservation of existing chiral centers) using MM94x force field and a Born Solvation model without cutoff constraints. All other parameters were left at default.

Crystal structures of HDAC2 (PDB code: 3max), HDAC4 (PDB code: 2vqm), HDAC7 (PDB code: 3c10), and HDAC8 (PDB code: 1t64) were retrieved from the protein data bank (PDB, http://www.ebi.ac.uk/pdbe/, accessed on 29 March 2022) and loaded into MOE. The Protonate3D functionality was applied to assign the correct ionization state and geometries to the protein atoms and to add hydrogen atoms. For the final docking, water molecules were discarded. Docking was performed using GOLD (version 4.1.2, The Cambridge Crystallographic Data Center, Cambridge, UK). No additional protein preparation was applied. Binding sites were defined by all residues within 5 Å distance from the corresponding ligands in the crystal structure. Docking was performed using GoldScore as scoring function. All other parameters were left at default. Docking was validated by comparing the highest-scoring docking pose of TSA to the pose of the ligand in the corresponding crystal structure of HDAC7 and HDAC8. In both cases, excellent overlays were obtained with root-mean-square deviation (RMSD) values below 1.5 Å. Docking poses of gossypol in the individual HDAC binding pockets were analyzed in MOE. To optimize the ligand–receptor interaction, energy minimization was applied using MM94x force field and a Born Solvation model without cutoff constraints.

### 4.4. HDAC Inhibition Profiling

The human HDAC profiling assay was performed on the basis of the Fluor de Lys^TM^ technology by Scottish Biomedical (Glasgow, UK) (as cell-free assay on protein level, with extracted HDACs). The percentage inhibition of the human HDACs 1 to 11 by gossypol was determined. Each value consists of two individual experiments, performed in duplicate. All assays were performed in 1% DMSO (final concentration).

### 4.5. Immunoblotting

HepG2 and HCT-116 cells (2 × 10^5^ cells/well) were seeded into 6-well plates and treated with gossypol the following day. After 6 h, the cells were harvested, washed once with phosphate-buffered saline (PBS), and resuspended in 100 µL lysing buffer (1% Nonident P40, 0.5 M Tris-Base (pH 7.6), 0.15 M NaCl). Lysates were stored at −80 °C, thawed, refrozen three times, and treated with a sonifier (60% output volume, 20 s). A volume of 50 µL with cellular protein was separated on 12% sodium dodecyl sulfate (SDS)-polyacrylamide gels under reducing conditions and transferred to polyvinylidene difluoride (PVDF) membranes (Hybond-P, Amersham Biosciences, Piscataway, NJ, USA). Membranes were blocked for 1 h in Tris-buffered saline (TBS; 150 mM NaCl, 13 mM Tris, pH 7.5) containing 5% nonfat dry milk powder. Next, the membranes were incubated with anti-vinculin (mouse, clone hVIN-1; CAS: V 9131; 1:5000; Sigma-Aldrich), anti-acetylated-tubulin (mouse, clone 6-11B-1; 1:2000; Sigma-Aldrich), polyclonal anti-histone H3 (rabbit, 1:10,000; Active Motif, Carlsbad, CA, USA), or anti-acetyl-histone H3 antibody (rabbit, CAS: 06-599; 1:5000; Merck Millipore, Billerica, MA, USA) overnight at 4 °C, then washed three times with TBS-T (TBS containing 0.02% Triton X-100), and incubated with peroxidase-conjugated goat anti-rabbit IgG (H + L) (1:8000; #170-6515, Bio-Rad Laboratories, Inc., Hercules, CA, USA) or goat anti-mouse IgG (H + L) antibody (1:4000; #170-6516, Bio Rad) for 45 min. Membranes were washed six times in TBS-T, and further detection was performed by the ECL Western blotting detection system on Hyperfilm-ECL (Amersham Biosciences, Piscataway, NJ, USA).

### 4.6. Real-Time Cell Proliferation Assay

HT-29 (10^4^ cells/well), HCT-116 (10^3^ cells/well), HepG2 (10^4^ cells/well), and Hep3B (5 × 10^3^ cells/well) were seeded in 96-well plates (E-Plate 96, Roche Applied Science, Mannheim, Germany). Real-time dynamic cell proliferation was monitored in 30 min intervals during 106 h using the xCELLigence system (Agilent Technologies Inc., Santa Clara, CA, USA). Cell index values were calculated using the RTCA Software Pro (2.3.2.) (Agilent Technologies Inc.). All curves were normalized at the beginning of the treatment period (10 h after seeding) applying the RTCA software [54].

### 4.7. Sulforhodamine B Assay

HT-29, HepG2, Hep3B, CCD 841 CoN, and CCD-18Co cells (5 × 10^4^ cells/well for 48 h and 2 × 10^4^ cells/well for 96 h), HCT-116 (2 × 10^4^ cells/well for 48 h and 1 × 10^4^ cells/well for 96 h) or PHHs (3 × 10^5^ cells/well) were seeded in 24-well plates. Growth inhibition was evaluated at dedicated time points by SRB assay [90]; data plotted represent the mean of optical density measurements related to untreated cells.

### 4.8. Analysis of sub2N Cell Population

HT-29, HCT-116, HepG2, and Hep3B (each 7.5 × 10^4^ cells/well) were seeded in 24-well plates and treated the next day. The cells were prepared for FACS analysis after incubation. In brief: supernatant, PBS after washing, and trypsinized cells were collected in a FACS tube, centrifuged at 250× *g* for 8 min, and resuspended in 200 µL FACS buffer (40 mg sodium citrate, 150 µL Triton X-100, 1 mg/mL propidium iodide, 5 mg ribonuclease A, 50 mL dd H_2_O). After incubation for 30 min on ice in the dark, cells were analyzed by flow cytometry (FACS Calibur, BD Biosciences, Heidelberg, Germany).

### 4.9. Caspase 3/7-Assay

HepG2 and HCT-116 cells (10^4^ cells/well) were plated in 96-well plates. Treatment was performed in 100 µL DMEM without FCS. The assay was performed as described by the manufacturer (Caspase Glo^®^ 3/7, Promega, Madison, WI, USA). In brief: after incubation, 100 µL reagent was added to each well, and plate was shaken for 5 min on an orbital shaker, following 55 min of incubation at RT. Afterward, luminescence was measured using ELISA reader (Tecan Genios Plus, Tecan Group Ltd., Männedorf, Switzerland). An amount of 5 µM STS was used as positive control.

### 4.10. ATP Assay

For this purpose, 1 × 10^5^ cells/well (HT-29, HCT-116, HepG2, and Hep3B), were seeded in 24-well plates and incubated overnight. After replacing the medium, cells were treated with different concentrations of gossypol and incubated for determined time points. Using the Roche ATP Bioluminescence Assay Kit CLS II, the ATP content was determined according to the manufacturer’s instructions. Luminescence was measured with the ELISA reader (Tecan Genios Plus; setting: integration time per well: 1000 ms, shake duration: 120 s, shake settle time: 480 s).

### 4.11. Aspartate Aminotransferase and Lactate Dehydrogenase Assay

PHHs were seeded in 24-well plates at a density of 3 × 10^5^ cells/well. After overnight incubation, cells were treated with different concentrations of gossypol and incubated for determined time points. AST concentration in the supernatant was analyzed with GOT-AST mono kit (Biocon™, Voehl/Marienhagen, Germany), as suggested by the manufacturer; all values refer to untreated control cells (% control). The photometric absorption at 340 nm was detected using the ELISA reader (GeniosPlus, Tecan) immediately after addition of the substrate (5 kinetic cycles, 120 s interval). LDH concentration in the supernatant was determined with LDH-P mono kit (Biocon™), as suggested by the manufacturer; all values refer to untreated control cells (% control). The photometric absorption at 340 nm was detected using the ELISA reader (Tecan Genios Plus,) immediately after addition of the substrate (5 kinetic cycles, 120 s interval).

### 4.12. Embryotoxicity Assay

Since chicken embryos were used in very early stages, no ethical approval was required, according to local animal care guidelines. Fertilized eggs of leghorn chickens (*Gallus gallus domesticus*) were obtained from a hatchery (Weiss, Kilchberg, Germany) and incubated at 38 °C in a temperature-controlled brooder (BRUJA Type 400a, Brutmaschinen Janeschitz, Hammelburg, Germany). For embryotoxicity testing, eggs were used after 50 h of incubation (equal to stage 13, according to Hamburger and Hamilton [91]), which corresponds to approximately six human gestational weeks [92]. The eggs were prepared as described previously [93]. Gossypol was applied in concentrations of 1 µM, 2.5 µM, 5 µM, 10 µM, 20 µM, and 50 µM (*n* = 8 each) on top of the blastoderm. The assay was performed in two different experiments on different experimental days. Eggs treated with solvent (ethanol) alone served as negative control. The eggs were sealed with adhesive tape and placed into the incubator. Then, 24 h, 48 h, and 72 h after the application of gossypol, viability of the embryos was determined by verification of heart action and blood flow in the chorioallantoic vessels. Survival rates were depicted as Kaplan–Meier curves.

### 4.13. Statistics

Statistical analysis was conducted with GraphPad Prism version 8.4 (GraphPad Software, San Diego, CA, USA). Statistical calculations for different assays were performed with one-way ANOVA and Dunnett’s multiple comparison test as post hoc test. According to the statistical evaluation, all gossypol treatment groups were compared vs. vehicle/control. *p*-values < 0.05 were considered statistically significant (* for *p* ≤ 0.05; ** *p* ≤ 0.01; *** *p* ≤ 0.001).

## 5. Conclusions

In summary, the data of the current study show that gossypol, a plant-derived and bioactive phenolic compound, acts as a pan-HDAC inhibitor at the epigenetic level and impacts cell viability by inducing apoptosis in both human malignant, as well as in non-malignant liver and colon cells and cell lines. Concentrations of gossypol ≥ 5 μM significantly limit the viability of non-malignant cells, and concentrations above 1 μM elicit embryotoxicity. Based on these results, the possible use of gossypol in human cancer therapy needs to be reconsidered, and both the risks and side effects of gossypol-based treatment should be re-evaluated. It might be promising to use gossypol not orally but as an external topical drug for the treatment of skin cancer or to develop derivatives with lower toxicity. However, these approaches have to be further investigated in detail.

## Figures and Tables

**Figure 1 pharmaceuticals-15-00438-f001:**
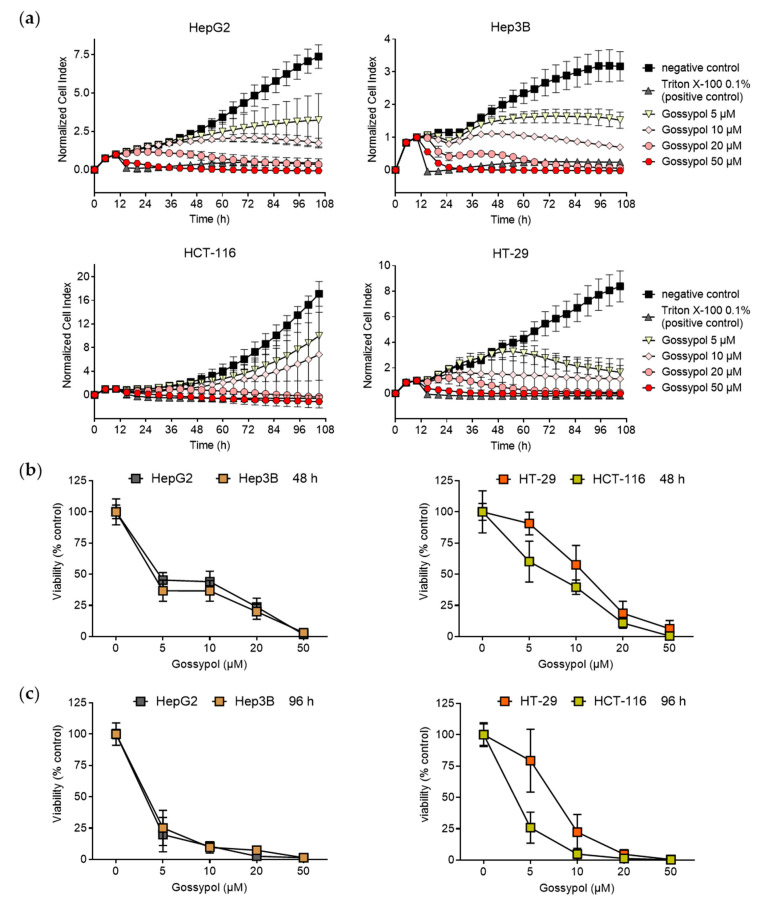
Reduced proliferation and viability of different hepatoma and colon carcinoma cell lines treated with gossypol. (**a**) Real-time proliferation curves. After 10 h incubation, cells were treated with different concentrations of gossypol (5 µM, 10 µM, 20 µM, 50 µM) or culture medium (negative control) and monitored for an additional 96 h. As control for complete inhibition of cell proliferation, 0.1% (*v*/*v*) Triton X-100 (positive control) was used. Cellular impedance was measured over the entire observation time using the xCELLigence^TM^ SP device (Agilent Technologies Inc., Santa Clara, CA, USA) and calculated by the RTCA Software Pro (2.3.2.). All CI values were normalized at 10 h before treatment started. Normalized CI values are displayed in intervals of 5 h. The mean values ± SD of three independent experiments are depicted. IC_50_ values for treatment duration of 72 h and 96 h were calculated with RTCA Software; HepG2, IC_50_ (72 h): 6.30 µM (SD = 1.73 µM), IC_50_ (96 h): 4.35 µM (SD = 2.10 µM); Hep3B, IC_50_ (72 h): 6.87 µM (SD = 1.02 µM), IC_50_ (96 h): 5.83 µM (SD = 0.81 µM); HCT-116, IC_50_ (72 h): 3.61 µM (SD = 1.54 µM), IC_50_ (96 h): 5.36 µM (SD = 2.87 µM), HT-29, IC_50_ (72 h): 11.8 µM (SD = 10.01 µM), IC_50_ (96 h): 14.0 µM (SD = 12.01 µM). Investigation of cell viability via SRB assay. Cells were treated for 48 h (**b**) and 96 h (**c**) with increasing concentrations of gossypol (5 µM, 10 µM, 20 µM, 50 µM), with culture medium (negative control) and 0.1% (*v*/*v*) Triton X-100 (positive control). Values represent means ± SD of three independent experiments, each performed in triplicate. CI, cell index; SRB, sulforhodamine B.

**Figure 2 pharmaceuticals-15-00438-f002:**
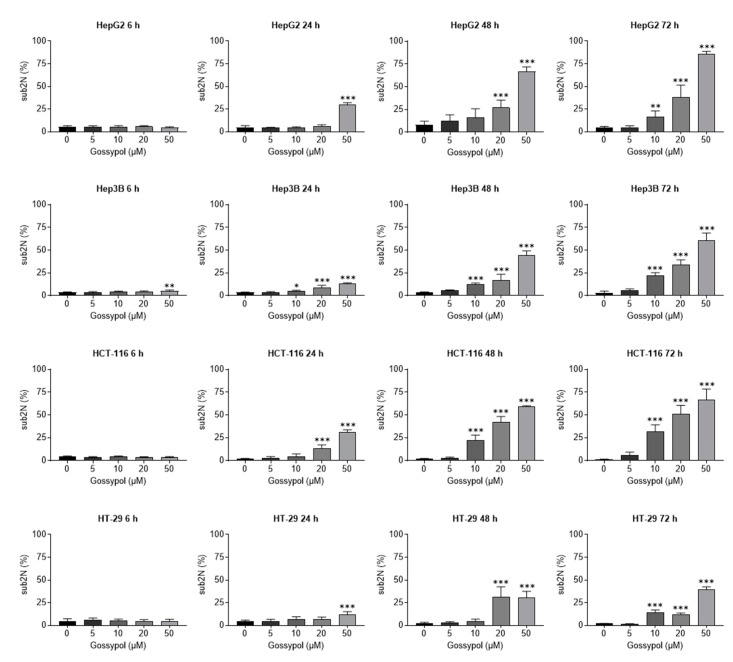
Gossypol induces apoptosis. A concentration-dependent (5 µM, 10 µM, 20 µM, 50 µM) increase in apoptotic cells by gossypol treatment was analyzed for hepatic and colon carcinoma cells (hepatoma: HepG2, Hep3B; colon carcinoma: HCT-116, HT-29). Detection of the apoptotic sub2N fraction was performed by flow cytometric analysis of PI-stained cells obtained after 6 h, 24 h, 48 h, and 72 h incubation with gossypol. Error bars represent mean ± SD of three independent experiments, each performed in triplicate. For statistical analysis, one-way ANOVA Dunnett’s multiple comparison test was applied. Confidence interval 95%; *: *p* ≤ 0.05; **: *p* ≤ 0.01; ***: *p* ≤ 0.001. PI, propidium iodide.

**Figure 3 pharmaceuticals-15-00438-f003:**
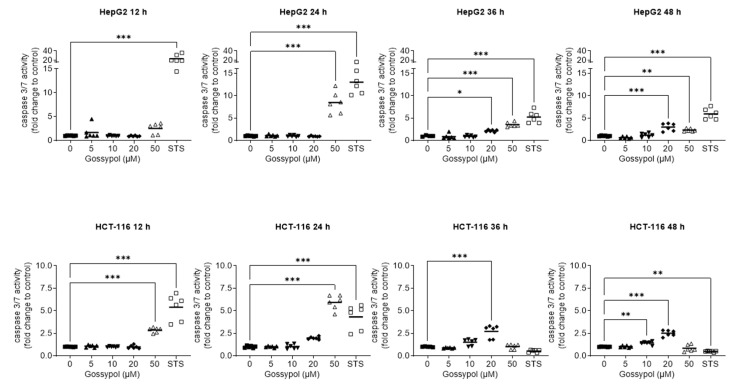
Gossypol activates caspases 3/7. Gossypol activates caspases in a concentration-dependent manner (5 µM, 10 µM, 20 µM, 50 µM,) in both p53 wt cancer cell lines HepG2 and HCT-116. Caspase activity was measured at four time points (12 h, 24 h, 36 h, 48 h). Untreated cells were used as negative control. STS (5 μM), an unspecific apoptosis inducer, was used as positive control. Bars represent mean of three independent experiments, each performed in duplicate, each spot corresponds to a measuring point (◼ = 0 µM gossypol; ▲ = 5 µM gossypol; ▼ = 10 µM gossypol; ◆ = 20 µM gossypol; △ = 50 µM gossypol; ◻ = 5 µM STS); statistical analysis with the Dunnett’s multiple comparison test, confidence interval 95%. *: *p* ≤ 0.05; **: *p* ≤ 0.01; ***: *p* ≤ 0.001. STS, staurosporine, wt, wild type.

**Figure 4 pharmaceuticals-15-00438-f004:**
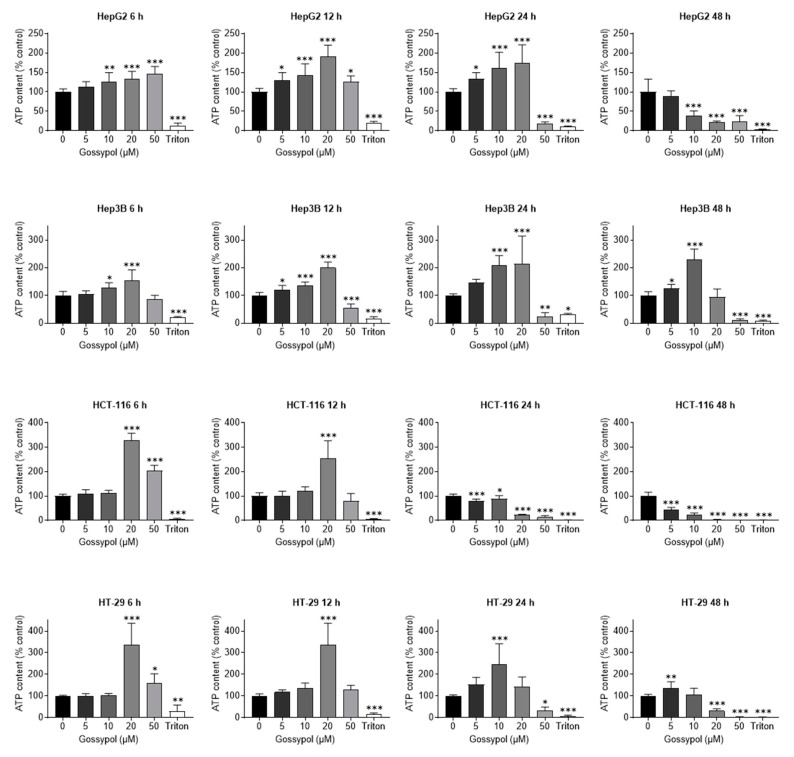
Gossypol alters cellular ATP levels. Photometric determination of the relative ATP content after gossypol incubation (6 h, 12 h, 24 h, 48 h) in a concentration-dependent manner (5 µM, 10 µM, 20 µM, 50 µM) for both cancer types (hepatoma: HepG2, Hep3B; colon carcinoma: HCT-116, HT-29). Medium alone (negative control) and 0.1% (*v*/*v*) Triton X-100 (positive control) were used for all cell lines. Error bars represent mean ± SD of three independent experiments, each performed in triplicate. For statistical analysis, one-way ANOVA with Dunnett’s multiple comparison test was applied. Confidence interval 95%; * *p* ≤ 0.05; ** *p* ≤ 0.01; *** *p* ≤ 0.001.

**Figure 5 pharmaceuticals-15-00438-f005:**
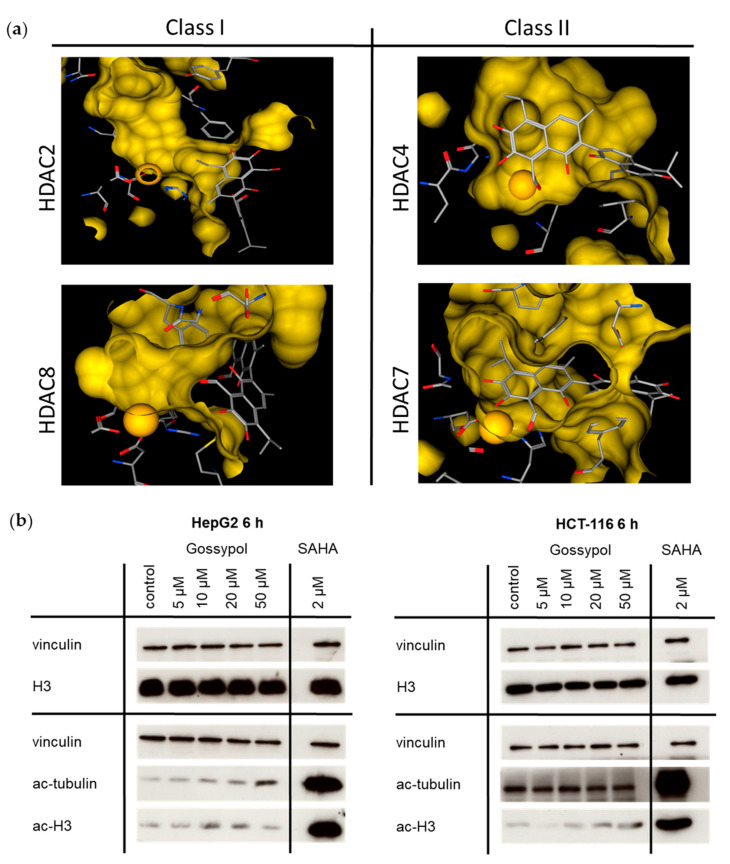
Gossypol inhibits HDAC activity. (**a**) Results of in silico docking analysis of gossypol with crystal structures of HDAC2 and 8 (class I, upper left, lower left) and HDAC4 and 7 (class II, upper right, lower right). The predicted binding modes of gossypol in the binding pocket of different HDACs are shown. The zinc ion (yellow sphere) and other residues of the catalytic center predicted to interact with gossypol are highlighted: carbon, gray; oxygen, red; nitrogen, blue. (**b**) Gossypol-related acetylation status. Western blot analysis of acetylated histone complex H3 (ac-H3) and acetylated tubulin (ac-tubulin) in HepG2 and HCT-116 tumor cells treated with 5 μM, 10 μM, 20 μM, or 50 μM of gossypol or solvent as control. Equal protein loading was verified by vinculin staining. As a reference and positive control for hyperacetylation, the cells were treated with 2 µM SAHA. Furthermore, as an additional control, Western blot analysis of total histone complex H3 (H3) was performed with the same lysates, and equal protein loading was again verified by vinculin staining (upper rows, left and right). HDAC, histone deacetylase; HDACi, histone deacetylase inhibitor; SAHA, suberoylanilide hydroxamic acid.

**Figure 6 pharmaceuticals-15-00438-f006:**
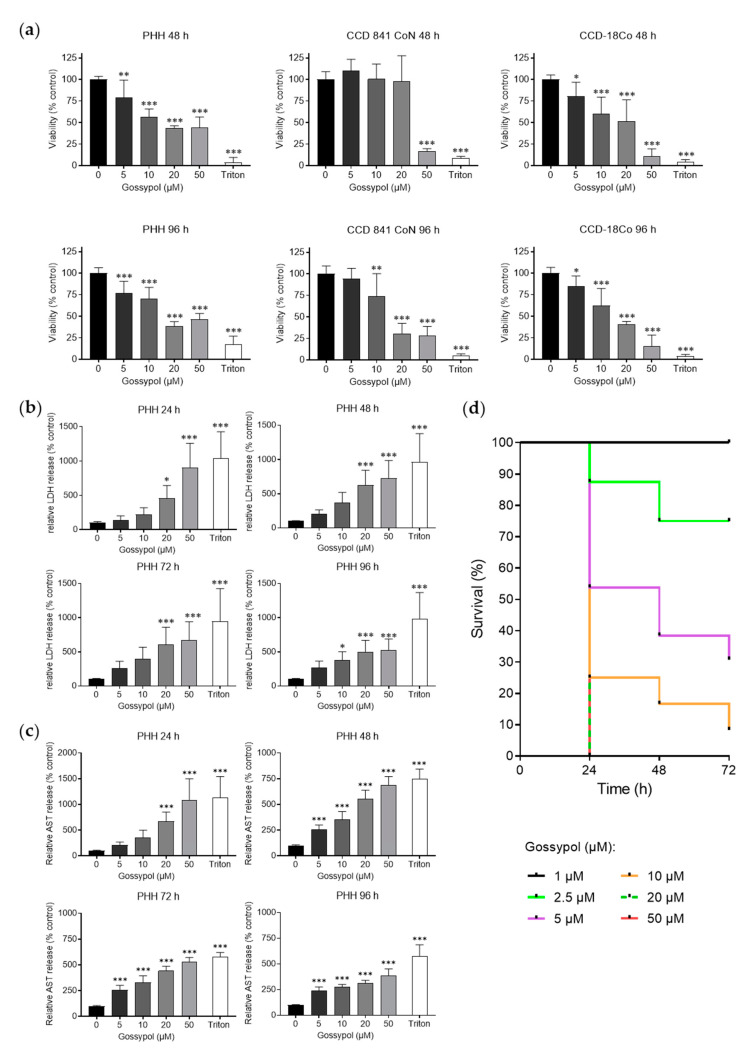
Gossypol induced toxicity. (**a**) Cell viability investigated via SRB assay with non-malignant PHHs from three different donors and epithelial non-malignant colon cells (CCD-18Co, CCD 841 CoN). Cells were treated for 48 h and 96 h with increasing concentrations of gossypol (5 µM, 10 µM, 20 µM, 50 µM), with culture medium as negative and Triton X-100 as positive controls. Values represent means ± SD of three independent experiments, each performed in triplicate. For statistical analysis, one-way ANOVA with Dunnett’s multiple comparison test was applied. Confidence interval 95%; * *p* ≤ 0.05; ** *p* ≤ 0.01; *** *p* ≤ 0.001. (**b**) LDH release of PHHs from three different donors into the supernatant was determined after 24 h, 48 h, 72 h, or 96 h treatment with increasing concentrations of gossypol (5 µM, 10 µM, 20 µM, 50 µM), with culture medium (negative control) and with Triton X-100 (positive control). For statistical analysis, one-way ANOVA with Dunnett’s multiple comparison test was applied. Confidence interval 95%; * *p* ≤ 0.05; *** *p* ≤ 0.001. (**c**) AST release of PHHs from three different donors into the supernatant was determined after 24 h, 48 h, 72 h, or 96 h treatment with increasing concentrations of gossypol (5 µM, 10 µM, 20 µM, 50 µM), with culture medium (negative control) and with Triton X-100 (positive control). For statistical analysis, one-way ANOVA with Dunnett’s multiple comparison test was applied. Confidence interval 95%; *** *p* ≤ 0.001. (**d**) Chicken embryotoxicity assay with different concentrations of gossypol (1 µM, 2.5 µM, 5 µM, 10 µM, 20 µM, and 50 µM). Chicken embryos stage 13 (corresponding to six human gestational weeks) were exposed to rising concentrations of gossypol to determine embryotoxic effects as LD_50_ value. Survival rates after 24 h, 48 h, and 72 h are depicted in a Kaplan–Meier plot. AST, aspartate aminotransferase; LDH, lactate dehydrogenase; PHH, primary human hepatocyte; SRB, sulforhodamine B.

**Table 1 pharmaceuticals-15-00438-t001:** Gossypol-induced inhibition of individual HDAC isoenzymes.

HDAC	% Inhibition (50 µM) ± SD
HDAC class I	
HDAC1	74.6 ± 15.5
HDAC2	77 ± 6.4
HDAC3	57.7 ± 9.6
HDAC8	82.7 ± 12.4
HDAC class II	
HDAC4	79.3 ± 4.1
HDAC5	65.9 ± 4.2
HDAC6	66.3 ± 11.9
HDAC7	77.2 ± 3.7
HDAC9	56 ± 13.9
HDAC10	80.6 ± 18.2
HDAC class IV	
HDAC11	74.3 ± 13.2

## Data Availability

Data is contained within the article.

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
