# Peer review of "Preclinical Efficacy and Toxicity Analysis of the Pan-Histone Deacetylase Inhibitor Gossypol for the Therapy of Colorectal Cancer or Hepatocellular Carcinoma"

_pharmaceuticals, 2022, doi:10.3390/ph15040438_

Round 1

Reviewer 1 Report

Materials

Source of Gossypol should be disclosed and described. Did the authors used R-(-)-enantiomer of gossypol, AT-101, or natural gossypol for this study?

Reference and Background

Background to mechanism of HDAC inhibitors in eliciting anti-tumour activities should expanded with addition references to be included. Here are some references that might be useful to discuss the apoptosis inducing activities of HDAC inhibitors and on the latest development of HDAC inhibitors as anti-cancer therapies.

  • Chüeh et al., ATF3 Repression of BCL-XLDetermines Apoptotic Sensitivity to HDAC Inhibitors across Tumor Types, Clinical cancer research, 2017
  • Chueh et al., Mechanisms of Histone Deacetylase Inhibitor-Regulated Gene Expression in Cancer Cells. Antioxidants & redox signalling, 2015
  • Wilson et al., Apoptotic sensitivity of colon cancer cells to histone deacetylase inhibitors is mediated by an Sp1/Sp3-activated transcriptional program involving immediate-early gene induction. Cancer research, 2010
  • Bondarev et al., Recent developments of HDAC inhibitors: Emerging indications and novel molecules. British journal of clinical pharmacology, 2021

HDAC inhibition

Mazzio et al., 2017 and Karaca et al., 2013 were two prior studies that demonstrated HDAC inhibitory activities elicited by gossypol using ELISA based assays. However, these studies failed to provide further evidence supporting the notion that gossypol is a genuine HDAC inhibitor (i.e. the prior studies only datapoints from 1 to 2 doses of gossypol used). The current study provided 2 lines evidence supporting gossypol as pan-HDAC inhibitor, demonstrated by hyperacetylation of histone H3 (Class I HDAC biomarker) using western blot analysis and inhibition of Class I, II and IV HDAC activity at a high gossypol dose (50 uM) by HDAC inhibition profiling (from Table 1). The claims could be strengthened by determining the IC50 of each HDACs by gossypol and additional western blot analysis of acetylated tubulin level (a direct substrate of HDAC6) following gossypol treatment. The induction of acetylated H3 by gossypol at 6 hr post treatment is significantly lower than SAHA treatment. Time-course experiments should be considered to determine whether this is due to a more slower acting nature of gossypol (a natural compound) in HDAC inhibition, in comparison to SAHA. Total Histone H3 should be used as positive control, instead of Vinculin. The loss of ac-H3 signal in HCT116 at 50 uM gossypol could be related to the loss of apoptotic cells. Both attached and floating cell populations should be collected and examined by ac-H3 signals and other assays used in this study. Further, the methods on HDAC inhibition profiling need to be expanded and further experimental details are required.

Gossypol induced toxicity (Figure 6)

The authors used non-malignant cell line models to demonstrate growth inhibitory effects of gossypol on PHH, CCD-18Co and CCD 841 CoN cells. Are these effects resulting from directly from on-target HDAC inhibition or off-target toxicity? Accompanying western blot analysis using antibodies recognising ac-H3 and ac-tubulin should be performed. Cleaved Caspase 3 western blot can also be performed to determine whether the effects are resulting from cell death or simply growth inhibition. Chicken embryo toxicity assays were performed, demonstrating in vivo toxicity resulting from gossypol treatment. Known HDAC inhibitors such as SAHA should be used controls for direct comparisons in chicken embryo toxicity assays and non-malignant cell line models, with accompanying ac-H3, ac-tubulin and cleaved Caspase 3 western blots. Together, these experiments will facilitate the further evaluation whether the toxicity is related to HDAC inhibition by gossypol or off-target toxicities.

Other comments:

Figure 1. IC50 should be estimated at 72 or 96 hr post drug treatment

Figure 2. cleaved Caspase 3 western blots should be performed.

Figure 3. Caspase 3/7 activity reduced at 36-48 hr for 50 uM. Are all live adherent and floating cells collected? 36-48 hr data may not be required.

In silico docking analysis is a good approach to study the binding of gossypol to HDAC enzymes. However, crystal structure is probably the gold standard. So, these data are interesting but is not necessarily required to be included as a the main figure.

Reviewer 2 Report

The authors assessed whether gossypol, a plant-derived compound, wolud be an HDAC inhibitor. For that, the authors evaluated the efficacy and toxicity of gossypol in colorectal and hepatocellular carcinoma cell lines. The paper is well written, the methods are appropriate and the results are novel. However, some major issues remain:

1 – Figure 1a is very difficult to interpret, because the symbols representing the different concentrations of gossypol are the same (triangles) with different colours. Unfortunately, the size of the figure makes it almost impossible to distinguish between the various conditions. I suggest that the authors use different symbols or increase the figure size. Also, authors should change the graph legends from “control” to “negative control” and “Triton X-100 0.1%” to “Triton X-100 0.1% (positive control)”. On line 192 (figure 1 legend) is said that Triton X-100 is the positive control for cell death, but the data represent proliferation curves, so this should be corrected.  

2 – On section 2.5, authors should indicate which cells were used to perform the in vitro analysis.

3 – Did the authors perform apoptosis assays on Hep3B and HT-29 cell lines? Since the authors clearly state that gossypol can induce apoptosis regardless of p53 status (first paragraph of section 3.1), it would be important to evaluate the effect of this compound also in cells with p53 gene deletion or mutation.

4 – On section 4.4, the authors stated that each value consisted of two individual experiments. Nonetheless, to indeed sustain their conclusion  each experiment should be performed at least 3 times.

5 – The Introduction and Discussion sections are too long. The authors should be more objective and compact both sections.

6 – Starting on line 178, the authors refer that HT-29 cells displayed decreased  cell viability after gossypol treatment. However, the indicated percentages are only valid for the 5 uM condition and not for both 5 uM and 10 uM, as stated. Please correct.

7 – In the legend of Figure 4, authors refered that Triton X-100 was used as a control for cell death for HepG2 cells. Why was it only used as a control for these cells? Which control was used for the other cells?

Minor issues:

Materials and Methods: line 551 (authors should add Germany as the country of origin of FCS), section 4.5 (antibody clones or references should be included, when available), line 648 (authors should add at least the country of origin of Tecan Genios Plus), line 663 (authors should add at least the country of origin of Biocon)

Typos: line 184 (it should be “p53-independent” instead of “p53 independent”), line 290 (“classical human” instead of “classical of human”), line 364/365 (“noted up to 50 uM” instead of “noted up to 20 uM”)

Reviewer 3 Report

The manuscript by Mayer and co-workers investigated the effects of gossypol on proliferation and apoptosis of hepatocellular and colorectal carcinoma cells. Based on in silico-screenings and subtype specific HDAC enzyme tests gossypol was identified as a pan-HDAC inhibitor with moderate HDAC-inhibitory efficacy. Antiproliferative and apoptosis-inducing effects of gossypol were seen both in p53 wildtype as well as in p53 mutated hepatocellular and colorectal carcinoma cell models. However, also pronounced cytotoxicity seemed to play a prominent role in the observed effects. Moreover, the cytotoxic effects were not restricted to cancer cells but were also seen in non-transformed hepatocytes, fibroblast cells and non-transformed intestinal epithelial cells. Furthermore, pronounced embryotoxicity was observed in chicken embryos.  

The authors concluded that gossypol should be regarded with caution or needs further optimization to become suitable for cancer treatment in the future

The major goal of the investigation is clear and generally the set of experiments is well chosen. However, there are some issues that need clarification before publication can be recommended:

- it is stated that HDAC overexpression/activity plays a prominent role in carcinogenesis of cancer in general. However, what is known about aberrant expression or dysregulated activity of HDAC in the two cancer cells models that were investigated here? What was the ratio to take HCC and CRC models for this investigation? Please explain?

- the authors conclude that “the pronounced cancer cell inhibitory capacity of gossypol was driven by its pan-HDACi activity inducing apoptosis”. I cannot follow this interpretation, because

1.) toxicity seems to be more responsible than HDAC inhibition or HDAC-inhibition induced apoptosis 2.) neither a dose-dependent increase in H3-actylation was observed in Hep-G2 or HCT-116 cells, which even showed a decline/absence in H3-acetylation after gossypol treatment. This does neither fit to the dose-dependent increase in apoptosis seen in both cell models and does also not fit to the general statement that gossypol acts as a pan-HDAC inhibitor.
3.) the experiments did not show a causative effect of HDAC (subtype) inhibition to the observed antiproliferative or apoptosis-inducing effects but only showed that apoptosis was induced by gossypol and that HDAC enzyme tests showed an inhibitory effect of gossypol in Hela extracts. If inhibition of nuclear HDACs was responsible than respective changes in gene expression should have been examined and if cytosolic HDAC activity (e.g. HDAC6) was responsible for the induction of apoptosis than for example respective siRNA silencing experiments should have been performed to mimic effects of HDAC-6 inhibition in the cancer cell models
4.) Why are no data given on the effects gossypol-induced apoptosis in non-transformed cells?

- the introduction is very detailed in terms of HDAC classes their inhibition by clinically relevant inhibitors etc., while interesting data on clinical trials of gossypol for cancer treatment is not mentioned at all. Since the compound is known for decades and is clinically investigated as an anticancer compound in a plethora of studies this should have been included here.

- Figure 1a: the figure is hardly legible, since the differently colored triangles for the different gossypol concentrations are too small. Please rearrange the chart and use symbols that can easily be distinguished.

-Figure 3: There is more or less no time-dependent or dose-dependent activation of the capsases 3/7 seen in the figure -except for the highest concentration of 50µM gossypol after 24h in HEP-&2 annd HCT-116 cells. Please check for significance of caspase 3/7 activation. If no significance is reached than the figure can either be reduced to showing 24h as an example for (non- significant) caspase activation only, or the figure can be skipped at all.      

- Figure 5c: How many independent repetitions of the H3-acetylation Western blot were performed? Is the calculation given for the exemplarily depicted blot, or does represent means +/-SD of all repetitions? Neither in materials and methods nor in the text any information on n is given.

- the number of references (119!) is far too high for an experimental paper. This is no review, so please cut down the number of references to approx. one third and cite actual literature, only. Please avoid including 20–30-year-old general literature on gossypol or HDAC research.

Round 2

Reviewer 3 Report

The authors sufficiently addressed all issues raised by reviewers. 

The revised manuscript can thus be now accepted for publication.